# Granulomatous Prostatitis, the Great Mimicker of Prostate Cancer: Can Multiparametric MRI Features Help in This Challenging Differential Diagnosis?

**DOI:** 10.3390/diagnostics12102302

**Published:** 2022-09-23

**Authors:** Elena Bertelli, Giulia Zantonelli, Alberto Cinelli, Sandro Pastacaldi, Simone Agostini, Emanuele Neri, Vittorio Miele

**Affiliations:** 1Department of Radiology, Careggi University Hospital, 50134 Florence, Italy; 2Department of Translational Research, Academic Radiology, University of Pisa, 56126 Pisa, Italy

**Keywords:** granulomatous prostatitis, prostate cancer, multiparametric MRI, diffusion-weighted imaging (DWI), dynamic contrast-enhanced MR imaging

## Abstract

Clinico-radiological presentation of granulomatous prostatitis (GP) is quite similar to cancer, and differential diagnosis can be very challenging. The study aims to highlight GP features based on clinical findings and multiparametric magnetic resonance imaging (mpMRI) characteristics. We retrospectively reviewed eleven patients from a cohort undergoing targeted biopsy between August 2019 and August 2021. Retrospective data including serum prostate-specific antigen (PSA) levels, PSA density and mpMRI findings were collected. Histopathology revealed seven cases of non-specific GP and four cases of specific GP as a result of intravesical Bacillus Calmette–Guérin (BCG) instillation. All lesions showed low signal intensity in T2w images, restricted diffusivity with hyperintensity in Diffusion-Weighted Imaging (DWI) and low Apparent Diffusion Coefficient (ADC) values. In Dynamic Contrast-Enhanced (DCE) imaging, the enhancement was high-peak and persistent in the majority of cases, especially in BCG-GPs. Moreover, almost all those latter lesions showed avascular core and peripheral rim enhancement. All areas identified on mpMRI were assessed with high to very high suspicion to hold prostate cancer (PIRADS v2.1 scores 4–5). Despite recent advances in imaging modalities and serological investigations, it is currently still a challenge to identify granulomatous prostatitis. Histopathology remains the gold standard in disease diagnosis. However, a differential diagnosis should be considered in patients with prior treatment with BCG.

## 1. Introduction

Granulomatous prostatitis (GP) is a relatively rare chronic inflammatory disease of the prostate, which can simulate prostate cancer (PCa). Actually, GP is a great tumor-mimicker: it has the same clinical and laboratory characteristics and imaging appearance of PCa. On digital rectal examination (DRE), there is an appreciable neoplastic-like nodular induration of the peripheral zone or a diffuse enlargement of the gland with consensual increase in serum Prostate-Specific Antigen (PSA) values, even if slight and transient [1]. Multiparametric magnetic resonance imaging (mpMRI) findings are equally not specific for this pathology since the most frequent presentation is represented by nodule/areas of signal hypointensity in T2w images and restriction of proton diffusivity (DWI) at the peripheral portion of the gland often without the involvement of the adipose plane, the finding of which is often indistinguishable from PCa [2].

Therefore, the diagnosis is obtained only by biopsy examination which allows demonstrating the characteristic inflammatory infiltrate with epithelioid cells, histiocytes and, occasionally, multinucleated giant cells, with or without evolution towards caseous necrosis [3].

Based on histopathological findings and probable etiology, Epstein and Hutchin classified GPs into the following types: idiopathic (non-specific GP, NGP), infectious (specific GP, SGP), iatrogenic (post-surgical), due to malacoplakia, associated with systemic granulomatous diseases or allergic diseases [4,5,6,7].

The differentiation between NGP and SGP is crucial because the first is self-limiting, whereas the latter needs specific treatment for causative factors [3].

The present work aims to describe the main clinico-laboratory and multiparametric-MRI characteristics in 11 patients with a bioptic diagnosis of GP.

## 2. Materials and Methods

The study was approved by our local institution review board and conducted in accordance with institutional guidelines, including the Declaration of Helsinki.

### 2.1. Population

This monocentric retrospective observational study included 563 consecutive patients who underwent MRI/US fusion-guided targeted biopsy based on a positive multiparametric MRI (mpMRI) result (PI-RADS v2.1 score ≥ 3) between August 2019 and August 2021 at the Uronephrologic Unit of our Radiology Department. Patients underwent mpMRI for clinical suspicion of PCa (DRE findings, total serum PSA > 4 ng/mL or > 2 ng/mL in patients with a family history, PSA velocity > 0.8 ng//mL year, Prostate Health Index ≥ 45). Retrospective data including age, serum prostate-specific antigen (PSA) levels and PSA density were collected. During MRI/US fusion-guided transperineally targeted biopsy, 3–5 targeted cores for each lesion were obtained in relation to their size. All patients also underwent 8–14 cores randomized standard prostate biopsy (6–10 samples for the peripheral zone and 2–4 samples for the transition zone) in relation to prostate volume.

### 2.2. mpMRI

All mpMRI were performed using a 1.5-tesla MR scanner equipped with an anterior pelvic phased-array 18-channel coil and a posterior spine phased-array 16-channel coil (Magnetom Aera, Siemens Medical Systems, Erlangen, Germany) at our Radiology Department.

The MRI PI-RADS v2.1 acquisition protocol included high-resolution T2w sequences in the axial (TR/TE = 4790/123 ms, voxel size = 0.3 × 0.3 × 3.0 mm^3^), sagittal (TR/TE = 4470/101 ms, voxel size = 0.3 × 0.3 × 3.0 mm^3^) and coronal (TR/TE = 3520/123 ms, voxel size = 0.3 × 0.3 × 3.0 mm^3^) planes, automatically interpolated from a voxel size of 0.74 × 0.63 × 3.00 mm^3^ by the MRI console; a T1-weighted sequence (TR/TE = 450/10 ms, voxel size = 0.6 × 0.6 × 3.0 mm^3^) in the axial plane; a multi-b DWI (b-values = [50, 100, 800, 1000] s/mm^2^, voxel size = 1.0 × 1.0 × 3.0 mm^3^, three directions) EPI sequence, automatically interpolated from a voxel size of 2.60 × 2.08 × 3.00 mm^3^ by the MRI console, whose corresponding ADC maps were automatically calculated using software on board of the MRI console; a high-b DWI (b-values: [1400, 1800] s/mm^2^, voxel size = 2.2 × 2.2 × 3.0 mm^3^, three directions) EPI sequence; a Dynamic Contrast Enhancement (DCE) assessment with time intensity curves evaluation, as previously described [8].

### 2.3. Image Analysis

All the mpMRI were evaluated by a radiologist dedicated to uronephrologic imaging with 25 years of experience.

Lesions were classified according to the PI-RADS v2.1 5-point scale [9].

### 2.4. Targeted Biopsy

All patients underwent free-hand transperineal MRI/US fusion-guided targeted biopsy with virtual navigation platform (MyLab™Twice Esaote, Genoa, Italy) at the Uronephrologic Unit of our Radiology Department.

## 3. Results

GP was histopathologically detected in 11/563 patients, with a global detection rate of 1.95%.

On pathological examination, seven (64%) had NGP and four (36%) had SGP as a result of intravesical Bacillus Calmette–Guérin (BCG) instillation for non-muscle invasive bladder cancer (BCG-induced GP, BCG-GP).

Patient characteristics and main mpMRI imaging features according to PIRADS v2.1 score are summarized in Table 1.

Mean patient age was 68 ± 8 (range 52–79) years. Mean PSA was 8.74 ± 6.7 ng/mL (range 1.5–17.29 ng/mL). High serum PSA levels were observed in eight/eleven patients (73%). Mean PSA density was 0.12 ± 0.11 ng/mL/cc (range 0.03–0.41).

Even though it was not possible to carry out statistical analysis due to the too small sample, we can notice a difference in PSA and PSA density values between the NGP and the BCG-GP group. Indeed, if we compare the two groups, we can note that in the SGP group, mean PSA was 12 ± 7.4 ng/mL and mean PSA density was 0.19 ± 0.15 ng/mL/cc; instead, in the NGP group, those values were significantly lower: 6.9 ± 6.1 ng/mL and 0.08 ± 0.05 ng/mL/cc, respectively.

### mpMRI Findings

Detailed mpMRI imaging features are shown in Table 2 (BCG-GP patients) and in Table 3 (NGP patients).

On mpMRI, eight/eleven patients had lesions with nodular morphology and an overall size between 1 and 3.5 cm (one of those had both a diffuse hypointense signal of the whole peripheral zone and a concomitant irregular, blurred pseudonodular lesion of the transition zone), two/eleven patients (case six and case nine) had a diffuse alteration of the whole or of a great part of the peripheral zone and one/eleven patients (case three) had a diffuse alteration of the whole gland associated to loss of normal peripheral zone/transition zone architecture.

In seven patients, the nodular lesions were confined to the peripheral zone (PZ), in one patient (case four), the lesion was located in the transition zone (TZ) and three patients (cases one, five and six) had lesions involving both the peripheral and transition zones.

Bulging and/or irregularity of the glandular capsule was observed in four patients (36%) and three of them had BCG-GP (75%). These findings could be suggestive of high-stage prostate cancer (≥T3a).

All lesions showed low signal intensity in T2w images and restricted diffusion in DWI with corresponding signal loss on ADC map. DCE imaging was available in all patients. Almost all the lesions showed hyperenhancement; only two NGP cases (cases four and nine) differed: the first was characterized by a transition large nodule (3.5-cm) with mild enhancement and the latter had a diffuse peripheral alteration with scarce enhancement. The enhancement was high-peak and persistent in all the BCG-GP cases and in one case (case five) of the NGP group. Moreover, the majority of the BCG-GP lesions showed the so-called “ring-enhancement” [10]: a peripheral rim enhancement, early and prolonged, and an avascular core (Figure 1).

Only one NGP case (case eleven) showed early enhancement and rapid contrast medium wash-out, which it is considered typical of aggressive prostate cancer [10].

All areas identified on mpMRI and subsequently proven to be GPs were classified as PIRADS v2.1 scores of four or five, that is high to very high suspicion to hold prostate cancer.

We present prominent imaging findings in four patients:

Case Three: A 70-year-old male presenting with irritative symptoms (frequency, urgency, hesitancy) and induration of the gland at DRE, PSA of 2 ng/mL, PSA density 0.03 ng/mL/cc. The mpMRI showed diffuse and inhomogeneous hypointensity on T2w images with loss of peripheral/transition zone differentiation, markedly restricted diffusivity and corresponding marked low ADC value, inhomogeneous hyperenhancement at DCE (Figure 2A–E). The PIRADS v2.1 score assigned was four. At the transrectal US (Figure 2F), preliminar to the fhTFTB (Figure 2G–g), there were multiple confluent hypoechoic areas both in the peripheral and transition zone. The histopathological analysis on the biopsy specimen showed NGP. The mpMRI (Figure 3A–D) performed 1 year later shows the almost complete resolution of the aforementioned findings.

Case Five: A 67-year-old patient with irritative symptoms (frequency and urgency), enlargement and induration of the prostate at DRE, PSA 11.4 ng/mL PSA density 0.09 ng/mL/cc. mpMRI (Figure 4) showed a 2.8-cm hypointense lesion of the left prostate base (arrow in Figure 4A), hyperintense in DWI (arrow in Figure 4B), with markedly low ADC value (arrow in Figure 4C) and an early and prolonged enhancement with high peak (arrow in Figure 4D). The PIRADS v2.1 score assigned was five. The patient also had a transition lesion on the prostate apex, not shown in Figure 4, characterized by a blurred low signal in T2w images, hyperintensity in DWI and corresponding marked low ADC value and early and prolonged enhancement with high peak too. The PIRADS v2.1 score assigned to the TZ lesion was five. At the transrectal US performed prior to the biopsy, we can see a wide hypoechoic peripheral area, which corresponds to the mpMRI PZ findings (arrow in Figure 4E). The histopathological analysis on the biopsy specimen showed NGP.

Case Six: A 52-year-old patient who performed intravesical BCG instillations for non-muscle invasive bladder cancer in the last 2 years. He presented with dysuria, positive DRE, PSA 22.9 ng/mL, PSA density 0.41 mg/mL/cc. mpMRI showed a blurred lesion in the transition zone and an extended peripheral alteration involving the left prostate lobe at the middle/base level with capsular bulging (Figure 5A). Both those alterations are hypointense in T2w images. In DWI, the peripheral alteration contained a small median area characterized by a markedly restricted diffusivity; also, the transition lesion is hyperintense in DWI (Figure 5B). Both findings have low ADC values, especially the peripheral one (Figure 5C). In DCE images, the peripheral lesion has a thick peripheral rim enhancement with an avascular core consistent with caseous necrosis and abscess formation (arrow in Figure 5D), whereas the transition lesion has an early and prolonged hyperenhancement (star in Figure 5D), due to the absence of central caseous necrosis. The PIRADS v2.1 score assigned was five for both lesions. The histopathological analysis on the biopsy specimen showed SGP, related to the BCG vesical instillation (BCG-GP).

Case Eight: A 62-year-old male with a history of high-grade bladder dysplasia treated with BCG and Pharmorubicin, nodular induration at DRE, PSA 3.3 ng/mL, PSA density 0.11 ng/mL/cc. In a context of a blurred hypointense alteration of the peripheral zone of the left middle lobe, in the T2w image, there was a markedly hypointense small nodule (arrow in Figure 6A), markedly restricted DWI (arrow in Figure 6B) and low ADC value (arrow in Figure 6C). In DCE image, the lesion shows a well-defined avascular core surrounded by an early rim enhancement (arrow in Figure 6D). The PIRADS v2.1 score assigned was five. The histopathological analysis on the biopsy specimen showed SGP, related to the BCG vesical instillation (BCG-GP).

In our series, the BCG-GP cases were characterized by high PSA and PSA density values, by low signal intensity lesions on T2w images, high signal intensity on DWI and markedly low ADC values; furthermore, 75% of BCG-GP cases had capsular bulging. All those findings could be suggestive for high-grade prostate cancer. Whereas, the enhancement pattern could be helpful in differentiating BCG-GP from cancer, according to the literature [10]. Indeed, in our series almost all the BCG-GP showed an early and prolonged enhancement of the peripheral rim and an avascular core, due to caseous necrosis and abscess formation. Instead, prostate cancer is usually characterized by early enhancement and rapid wash-out of the contrast medium. Although this characteristic imaging feature, we performed biopsy in all patients to confirm the clinical suspicion and to exclude concomitant prostate cancer.

## 4. Discussion

Granulomatous prostatitis is an unusual benign inflammatory condition of the prostate. This benign pathology is still considered rare although its incidence is rising due to the extensive use of BCG and to the increase in TURP and prostate biopsies [11]. According to the classification compiled by Epstein and Hutchin in 1984, GP may be idiopathic, i.e., non-specific or secondary to causes including specific infections, Calmette–Guérin bacillus therapy, surgery and, more rarely, systemic causes [12]. Its incidence is estimated at 3.3% in all prostatic inflammatory lesions [13]. In the present study, the incidence of GP non-associated to PCa in our cohort of men undergoing MRI/US fusion prostate biopsy was 1.95%, which was higher than those reported by Kumbar et al. and by Oppenheimer et al. [3,14] and in accordance with those reported by Shukla et al. and by Lee et al. [2,11]. This higher incidence compared to less recent works could be due to differences in patient populations. In the studies by Oppenheimer et al. and by Kumbar et al., patients underwent transrectal biopsy for raised PSA and/or abnormal DRE or TRUS findings. In the more recent works, as suggested in the literature [15,16] and in the latest EAU Guidelines [17], patients with those characteristics underwent mpMRI prior to biopsy. The biopsy is based on a positive mpMRI, thus avoiding unnecessary biopsies. Furthermore, in our series we performed a specific type of targeted biopsy focused on the MRI findings, the free-hand transperineal MRI/US fusion-guided targeted biopsy. This targeted biopsy enables real time co-registration of MRI and US images via software platforms (Figure 2G-g), so we can perform an accurate sampling of the lesion detected on the mpMRI. The targeted biopsy approach permits a better selection of patient population, enriching the cohort of patients with abnormal findings. In recent years some authors have proposed also some nomograms to predict the risk of PCA in patients with elevated PSA and indeterminate MRI results (i.e., PIRADS 3) [18].

NGP is the most common etiology among GPs, accounting for about 75% [14]. In the present study, it was also the most common entity diagnosed, with an incidence of 64%. The etiology remains unclear although repeated urinary tract infections, surgery (TURP, open prostatectomy) and bladder instillation of the BCG have been mentioned in the literature as contributing causes [3,4,19]. At the heart of the etiopathological process, it is a possible obstruction of the prostatic ducts with consequent stasis of secretions and rupture of the epithelium. Consequently, the leakage into the surrounding stroma of debris and secretions leads to a localized inflammatory reaction [20].

The clinical, laboratory and radiological presentations of granulomatous prostatitis, especially of BCG-GP, are similar to cancer, and differential diagnosis can be very difficult. Surely, it is important to accurately investigate the medical history to rule out previous intravesical BCG instillation.

Clinically granulomatous prostatitis can lead to increased PSA levels. In this study, PSA values ≥ 4 ng/mL (threshold value) were found in eight cases (73%). Similarly, PSA density was high (≥0.08 ng/mL/cc) in seven/eleven patients (64%). Interestingly, all BCG-GP patients had positive DRE (nodular induration) and higher PSA and PSA density values (12 ± 7.4 ng/mL and 0.19 ± 0.15 ng/mL/cc, respectively) compared to NGP ones (6.9 ± 6.1 ng/mL and 0.08 ± 0.05 ng/mL/cc, respectively). Even if these findings are not statistically significant due to our too small sample, it is interesting to confirm that BCG-GP is a great PCa mimicker not only for the radiologist, but also for the clinicians.

Radiologically, GP on T2w images is characterized by a low signal intensity, a diffuse or nodular alteration usually with blurred margins and could be associated to capsular bulging or irregularity [2]. The high cell density results in restricted diffusivity and corresponding low ADC values. Areas of inflammation can increase perfusion on DCE sequences giving rise to a false-positive result. As reported in [20], granulomatous prostatitis can have two main appearances in MRI: tumor-like or abscess-like. In the first case, the most frequent, it can be difficult to distinguish from prostate cancer, with signal hypointensity in T2w, restriction of diffusion due to high cell density and low ADC values. Several authors have shown that the median ADC value of granulomatous non-necrotic foci was lower than those reported for high-grade PCa [2,20]. The abscess-like appearance is reported as rare and it corresponds to a caseous abscess induced by a severe caseous necrotic process. MRI findings include a moderate signal hypointensity in T2w, a complete absence of enhancement due to necrosis associated with focal hyperintensity on high b-value DWI images and low ADC value.

Limited studies have investigated the appearances of GP on DCE imaging [10,21]. Kawada et al. found that early and prolonged ring contrast enhancement on DCE sequences corresponded to caseous necrosis in BCG-induced GP. Similar to the aforementioned findings, in our study, almost all the BCG-GP cases showed an early and prolonged peripheral rim enhancement and an avascular core in the DCE sequences, due to caseous necrosis and abscessualization.

Although some lesions, particularly those induced by BCG, may show a decrease in size and even a return to the normal appearance of the prostate gland with no detectable residual cavity on repeat mpMRI after one year of anti-tuberculosis treatment [10,21], in our study all patients underwent transperineal prostate biopsy. This choice is due, nowadays, to the lack of certain and definite imaging findings and to the need to rule out concomitant prostate cancer. Studies on larger series or, maybe, the use of artificial intelligence methods [8] such as radiomic features [22] and, in particular, deep learning methods, could help the radiologist to unravel this challenging differential diagnosis.

## 5. Conclusions

The differential diagnosis between GP and PCa on MR imaging remains challenging. These two entities present an overlap of both clinical and imaging features. For this reason, all patients with suspected prostatic lesions on mpMRI require, nowadays, histopathological confirmation of these areas with targeted biopsy.

## Figures and Tables

**Figure 1 diagnostics-12-02302-f001:**
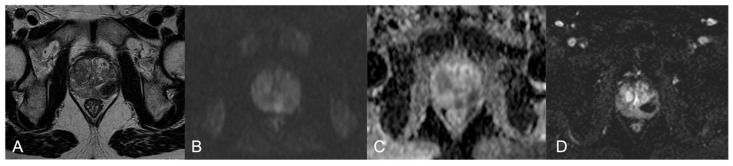
BCG-GP in a 62-year-old patient (case 1). MpMRI shows a 2-cm hypointense nodule in the left-middle lobe (**A**), predominantly isointense in DWI with a slightly hyperintense rim (**B**), marked low ADC value (**C**) and peripheral rim enhancement with a wide avascular core (**D**). PIRADS v2.1 4 PZpm mid-left.

**Figure 2 diagnostics-12-02302-f002:**
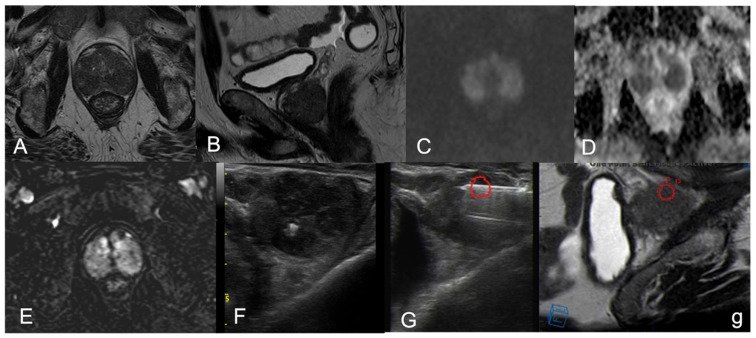
Non-specific GP in a 70-year-old patient (case 3). The mpMRI shows diffuse and inhomogeneous hypointensity on T2w axial (**A**) and sagittal (**B**) images with loss of peripheral/transition zone differentiation, markedly restricted diffusivity at high b-value (**C**) and corresponding marked low ADC value (**D**), inhomogeneous hyperenhancement at DCE (**E**). PIRADS v2.1 score 4 PZ diffuse. At transrectal US (**F**), preliminar to the fhTFTB, we can see multiple confluent hypoechoic areas both in the peripheral and transition zone. The free-hand transperineal fusion MRI/US-targeted biopsy (**G**,**g**): on the US image in (**G**) we can appreciate the needle entering into the target in red, in (**g**) we have the corresponding MRI-guidance image with the same target in red.

**Figure 3 diagnostics-12-02302-f003:**
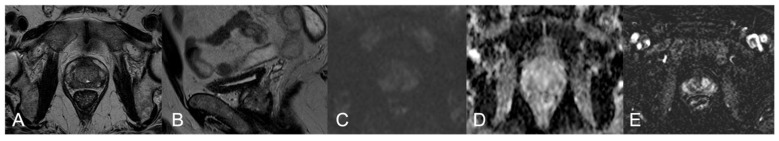
The mpMRI ((**A**): axial T2w image; (**B**): sagittal T2w image; (**C**): high b-value DWI; (**D**): ADC map; (**E**): DCE image) performed 1 year later shows the almost complete resolution of the findings mentioned in Figure 2 and in the text (case 3).

**Figure 4 diagnostics-12-02302-f004:**
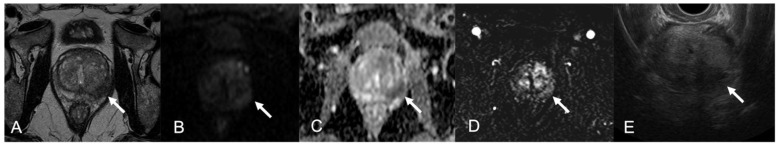
Non-specific GP in a 67-year-old patient (case 5). mpMRI showed a 2.8-cm hypointense lesion of the left prostate base (arrow in (**A**)), hyperintense in DWI (arrow in (**B**)) with markedly low ADC value (arrow in (**C**)) and an early and prolonged enhancement with high peak (arrow in (**D**)). At the transrectal US performed prior to the biopsy, we can see a wide hypoechoic peripheral area, which corresponds to the mpMRI findings (arrow in (**E**)). PIRADS v2.1 5 PZpl mid-left.

**Figure 5 diagnostics-12-02302-f005:**
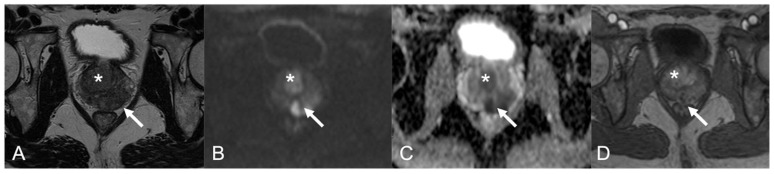
BCG-GP in a 52-year-old patient (case 6). In the T2w image, we can see a lesion in the transition zone (star in (**A**)) and an extended peripheral alteration involving the left prostate lobe at the middle/base level (arrow in (**A**)) with capsular bulging. In DWI, there is a median area characterized by a markedly restricted diffusivity (arrow in (**B**)); also, the transition lesion is hyperintense in DWI (star in (**B**)). The 2 findings (star and arrow in C) have low ADC value, especially the peripheral one (arrow in (**C**)). In DCE image, the peripheral lesion has a thick peripheral rim enhancement with an avascular core due to caseous necrosis (arrow in (**D**)), whereas the transition lesion has an early and prolonged hyperenhancement (star in (**D**)), due to the absence of central caseous necrosis. PIRADS v2.1 5 PZpm mid-base left (arrows) and 5 Tza mid-base right (stars).

**Figure 6 diagnostics-12-02302-f006:**
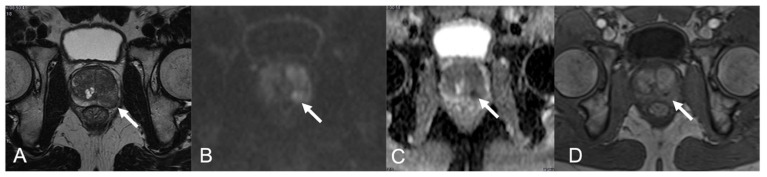
BCG-GP in a 62-year-old patient (case 8). MpMRI shows diffuse alteration of the left middle lobe peripheral zone with capsular bulging (arrow in (**A**)), with a marked hypointense 1-cm nodule markedly hyperintense on DWI (arrow in (**B**)), with marked low ADC value (arrow in (**C**)) and with an early and persistent peripheral rim enhancement and avascular core (arrow in (**D**)). PIRADS v2.1 5 PZpm mid-base left.

**Table 1 diagnostics-12-02302-t001:** Clinical parameters and main MR Imaging Features of GPs in 11 Patients.

Case	Age (years)	PSA Levels (ng/mL)	PSA Density (ng/mL/cc)	Volume (cc)	Location	T2WI Score	DWI Score	DCE Score	PIRADS v2.1	Hystology
1	62	10	0.19	54	PZ; TZ	4; 5	4; 3	+; +	4; 5	BCG-GP
2	72	7.96	0.08	95	PZ	4	4	+	4	BCG-GP
3	70	2	0.03	58	PZ diffuse	5	5	+	5	NGP
4	69	17.29	0.16	110	TZ	5	5	+	5	NGP
5	67	11.4	0.09	121	PZ; TZ	5; 5	5; 5	+; +	5; 5	NGP
6	52	22.9	0.41	55	PZ diffuse; TZ	5; 5	5; 5	+; +	5; 5	BCG-GP
7	79	2.79	0.07	40	PZ	3	3	+	4	NGP
8	62	7	0.11	65	PZ	4	5	+	5	BCG-GP
9	78	1.5	0.04	38	PZ diffuse	3	4	-	4	NGP
10	68	10.09	0.13	80	PZ	4	4	+	4	NGP
11	67	3.3	0.03	96	PZ	4	3	+	4	NGP

PSA = Prostate-Specific Antigen, TZ = Transition Zone, PZ = Peripheral Zone, T2WI = *T*2-weighted imaging, DWI = diffusion-weighted imaging, DCE = dynamic contrast-enhanced, PI-RADS = Prostate Imaging Reporting and Data System, NGP = Non-specific Granulomatous Prostatitis, BCG-GP = Bacillus Calmette–Guérin-induced Granulomatous Prostatitis.

**Table 2 diagnostics-12-02302-t002:** mpMRI imaging features in BCG-GP patients.

Case n.	Location	T2	DWI	ADC	DCE	PIRADS
1	PZpm mid-left; TZa mid-base right	2-cm hypointense PZ nodule; 1.5-cm marker hypointense irregular TZ lesion	Isointense nodule; Slightly hyperintense	Marked low ADC value; Low ADC value	Peripheral rim enhancement and avascular core; Inhomogeneous, early and prolonged hyperenhancement	4; 5
2	PZpl mid-left	2.5-cm hypointense nodule, capsular bulging	Hyperintense	Low ADC value	Thick peripheral rim enhancement, avascular core	4
6	PZ diffuse (index lesion PZpm mid-base left; TZa base right	Confluent pseudonodular hypointense areas, capsular bulging; 2-cm marked hypointense irregular TZ lesion	Marked hyperintense; hyperintense	Marked low ADC value; Marked low ADC value	Peripheral rim enhancement and avascular core in the index lesion; inhomogeneous, intense and prolonged enhancement	5; 5
8	PZpm mid-base left	Markedly hypointense nodule, focal capsular bulging	Marked hyperintense	Marked low ADC value	Peripheral rim enhancement, avascular core	5

**Table 3 diagnostics-12-02302-t003:** mpMRI imaging features in BCG-GP patients.

Case n.	Location	T2	DWI	ADC	DCE	PIRADS
3	PZ and TZ diffuse	Diffuse, inhomogeneous hypointensity, poor differentiation between PZ and TZ, glandular capsule preserved	Marked hyperintensity	Marked low ADC value	Inhomogeneous hyperenhancement, high peak	5
4	TZp mid-base right	3.5-cm hypointense blurred nodule	Hyperintense	Marked low ADC value	Mild enhancement	5
5	PZpl mid-left; TZa mid-left	2.8-cm hypointense area, capsular bulging; irregular, blurred marked hypointense TZ lesion	Hyperintense; hyperintense	Marked low ADC value; Marked low ADC value	Early and prolonged enhancement, high peak	5; 5
7	PZa right	0.9-cm Hypointense nodule	Mild hyperintense	Low ADC value	Hyperenhancement	4
9	PZ diffuse	Blurred and confluent mild hypointense areas	Hyperintense	Low ADC value	Scarce enhancement	4
10	PZpm right	1.3-cm mild hypointense nodule	Hyperintense	Low ADC value	Early enhancement, high peak	4
11	PZpm mid-left	Hypointense nodule	Mild hyperintense	Low ADC value	High and early enhancement, high peak, early wash-out	4

## Data Availability

Not applicable.

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
