# Peer review of "Granulomatous Prostatitis, the Great Mimicker of Prostate Cancer: Can Multiparametric MRI Features Help in This Challenging Differential Diagnosis?"

_diagnostics, 2022, doi:10.3390/diagnostics12102302_

Round 1

Reviewer 1 Report

Dear Editor and Authors,

The most important limitation of the study is the insufficient sample size and the lack of statistical analysis. However, the strength of the study is that the cases are summarized one by one and very well. With this aspect, it can be considered as a good descriptive study. However, some revisions are necessary for the further improvement of the article.

Abstract

-          Give the clear form of ADC, DWI, DCE, etc. before the first use.

Introduction

-          “PSA (Prostate Specific Antigen)” should be revised as “Prostate Specific Antigen (PSA)”.

Material and methods

-          Do you follow DRE, PSA, PSA velocity in the radiology outpatient clinic and give a biopsy indication? Why is there no urologist among authors?

-          In the study, MpMRI indications do not comply with uro-oncology guidelines. How do you explain this?

Results

-          Ok.

Discussion

-          It may be possible to predict PCA using some nomograms. Add this to the discussion section as a short paragraph. Please cite the article with doi: 10.1111/ajco.13347.

Author Response

Dear Editor and Authors,

The most important limitation of the study is the insufficient sample size and the lack of statistical analysis. However, the strength of the study is that the cases are summarized one by one and very well. With this aspect, it can be considered as a good descriptive study. However, some revisions are necessary for the further improvement of the article.

Thank you for your in-depth review and valuable comments, which have helped us to improve the quality and clarity of the report. We greatly appreciate your suggestions.

Point 1:       Abstract   Give the clear form of ADC, DWI, DCE, etc. before the first use.

Response 1: We felt so sorry for our negligence. We have revised our manuscript accordingly.

Point 2: Introduction  “PSA (Prostate Specific Antigen)” should be revised as “Prostate Specific Antigen (PSA)”.

Response 2: Thank you for your kind suggestion. We have corrected the sentence as you have indicated.

Point 3: Material and methods Do you follow DRE, PSA, PSA velocity in the radiology outpatient clinic and give a biopsy indication? Why is there no urologist among authors?

Response 3: Thanks for your kind suggestions, which are valuable for improving the accuracy of the manuscript. We were also concerned about this issue. The indication for the biopsy is always given by the urologist. When we perform the biopsy we do the DRE prior to the transrectal US as well, and we take a look to PSA and PSA velocity, also because we report these values in the form we fill in for the histopathological analysis. There are no urologists among authors because this is a purely radiological paper (we follow an internal institutional policy).

Point 4: In the study, MpMRI indications do not comply with uro-oncology guidelines. How do you explain this?

Response 4: We greatly appreciate you for this in-depth and constructive comment.

At our University-Hospital we have a dedicated Prostate Unit, and our urologists have validated an internal pathway for patients with suspected prostate cancer, based on the latest Guidelines and on their experience (we are a tertiary care University Hospital and our Urology Department is one of the biggest and most attractive of our country, Italy). The mpMRI indications written in our work are those written by our Urologists in the pathway abovementioned.

Point 5: Discussion It may be possible to predict PCA using some nomograms. Add this to the discussion section as a short paragraph. Please cite the article with doi: 10.1111/ajco.13347.

Response 5: Thank you for you valuable comment, which helped us tremendously to improve the quality of our work. We added a short paragraph in the revised version of the manuscript.

Reviewer 2 Report

This is the case series study to describe the MRI characteristics of granulomatous prostatitis (GP). granulomatous prostatitis is a relatively rare chronic inflammatory disease of the prostate. It is usually indolent but sometimes mimics prostate cancer. They presented 11 cases of GP and showed that the MRI findings were quite similar to that of prostate cancer. They concluded that a biopsy is necessary for differential diagnosis.

Since little has been known about the MRI findings of GP, the information provided in this study is novel and important. The manuscript was well-written and I have only a few suggestions.

1. Did the patients undergo only target biopsy? or they also received systemic biopsy? If a systemic biopsy was performed, is there any case of GP with normal MRI findings? 

2. For patients who were diagnosed as BCG-induced GP, did they receive any anti-tubercular treatment? I also wonder whether all SGPs really need treatments or not (line 48).

3. (line 91) 1,95%. -> 1.95%

Author Response

This is the case series study to describe the MRI characteristics of granulomatous prostatitis (GP). granulomatous prostatitis is a relatively rare chronic inflammatory disease of the prostate. It is usually indolent but sometimes mimics prostate cancer. They presented 11 cases of GP and showed that the MRI findings were quite similar to that of prostate cancer. They concluded that a biopsy is necessary for differential diagnosis.

Since little has been known about the MRI findings of GP, the information provided in this study is novel and important. The manuscript was well-written and I have only a few suggestions.

We would like to thanks the Reviewer for His/Her nice and valuable suggestions.

Point 1: Did the patients undergo only target biopsy? or they also received systemic biopsy? If a systemic biopsy was performed, is there any case of GP with normal MRI findings? 

Response 1: Thanks for your kind suggestions, which are valuable for improving the accuracy of the manuscript. We performed also a randomized biopsy, but we didn’t find any case of GP with completely normal MRI findings, even in the randomized samples.

We were concerned about this issue, and have added the sentences about targeted and randomized biopsy in the revised version.

Point 2: For patients who were diagnosed as BCG-induced GP, did they receive any anti-tubercular treatment? I also wonder whether all SGPs really need treatments or not (line 48).

Response 2: Thank you for your interest and constructive question. All the 4 patients affected by BCG-induced GP received antitubercolous treatment. So in our series all the SGPs needed treatment.

Point 3: (line 91) 1,95%. -> 1.95%

Response 3: We felt so sorry for our negligence. We have corrected our manuscript accordingly (line 89 in the revised manuscript).